# Acute Exposure to Ozone Affects Circulating Estradiol Levels and Gonadotropin Gene Expression in Female Mice

**DOI:** 10.3390/ijerph22020222

**Published:** 2025-02-05

**Authors:** Dustin Rousselle, Patricia Silveyra

**Affiliations:** 1Department of Environmental and Occupational Health, School of Public Health, Indiana University, Bloomington, IN 47405, USA; drousse@iu.edu; 2Department of Medicine, Indiana University School of Medicine, Indianapolis, IN 46202, USA

**Keywords:** ozone, sex hormones, air pollution, estradiol, folliculogenesis, HPG axis

## Abstract

Ozone, a critical air pollutant, has been shown to lead to systemic inflammation that can alter bodily functions, including hormone secretion, fertility, and the hypothalamic–pituitary–gonadal (HPG) axis. This study aimed to quantify changes in hormone production and follicle development after acute exposure to ozone using an animal model to identify the potential mechanisms underlying the observed effects of air pollution exposures on fertility and hormone secretion. To accomplish this, regularly cycling 8-week-old female C57BL/6J mice were exposed to 2 ppm of ozone or filtered air (control) for 3 h on the day of proestrus. Blood, ovaries, brain tissues, and pituitary glands were collected at 4 h after exposure to evaluate hormone levels, ovarian follicle distribution, and gene expression. Ovaries were also harvested at 24 h post-exposure. We found that at 4 h after ozone exposure, mice had significantly higher (30%) circulating estradiol levels than mice exposed to filtered air. This effect was accompanied by a decrease in mRNA expression of gonadotropin genes (LH, FSH) and gonadotropin-releasing hormone in the pituitary gland. Analysis of ovarian tissue at 4 h and 24 h after exposure showed no significant changes in follicle composition or the expression of steroidogenesis genes. We conclude that acute ozone exposure affects sex hormone levels and disrupts the HPG axis. Future studies addressing chronic or long-term effects of air pollution exposure are needed to elucidate the mechanisms by which ambient ozone affects endocrine function.

## 1. Introduction

Ozone is a highly reactive gas known to cause lung inflammation, oxidative stress, and other deleterious health effects when inhaled [1,2]. In addition to lung effects, ozone exerts systemic inflammatory responses affecting different organs and systems [3,4]. These systemic effects are caused by the likely release of inflammatory biomarkers into the circulation that may affect systems such as the hypothalamic–pituitary–gonadal (HPG) axis [5]. 

In females, the HPG axis regulates hormones, including luteinizing hormone (LH), follicle-stimulating hormone (FSH), estradiol, and progesterone. The hypothalamus releases gonadotropin-releasing hormone (GnRH), which prompts the pituitary gland to release both LH and FSH [6]. In turn, LH prompts the theca cells of the preovulatory follicles to produce androgens, and FSH prompts the granulosa cells to produce aromatase to convert the androgens into estrogens [7]. These hormones are also responsible for the development and maturation of follicles in the ovaries; FSH stimulates growth and the expression of the LH receptor in early follicles, and a surge of LH initiates the release of the mature oocyte [8].

Follicles develop through several stages. First, an oocyte exists as a primordial follicle before becoming a primary follicle after activation. The primary follicles develop into secondary follicles as the simple single layer of granulosa cells becomes stratified. Secondary follicles develop into antral or preovulatory follicles when they acquire an antrum (a fluid-filled space within the follicle). Ovulation occurs when the mature oocyte is released from the follicle for fertilization. The remaining follicle becomes a corpus luteum, which releases progesterone and estrogen to help with the implantation and zygote development [9]. Most follicles will undergo atresia, in which the developing follicle breaks down and becomes non-viable to save energy for the leading developing follicles [10].

Unlike the 28-day human menstrual cycle, mice undergo a similar cycle of hormones over 4 days, called the estrus cycle. The four stages of the cycle are proestrus, estrus, metestrus, and diestrus. In proestrus, a surge in estradiol and LH precedes ovulation and the estrus phase [11]. Similarly, folliculogenesis follows a similar pattern to that of humans but in a shorter timeframe.

Epidemiological studies have suggested that exposure to air pollutants like ozone can alter hormone secretion, affect fertility, and result in premature birth [12,13,14,15]. Despite these observations, no recent work has been conducted to study the effects of ozone in an experimental study, resulting in the mechanisms responsible for these associations not being elucidated. In the current study, we use an animal model, previously used to understand the effects of ozone on respiratory health [16,17], to investigate the effects of ozone exposure on the HPG axis, including hormone secretion, folliculogenesis, and genes related to steroidogenesis. We hypothesize that acute ozone exposure alters the development of ovarian follicles and affects ovarian hormone secretion.

## 2. Materials and Methods

### 2.1. Animal Treatment

Adult (8-week-old) female C57BL/6J mice were purchased from The Jackson Laboratory and maintained on a 12 h day–night cycle. The stages of the estrous cycle were determined every morning by vaginal smear analysis for at least 3 consecutive cycles, as carried out by us previously [18]. On the morning (11 a.m.) of proestrus, mice were exposed to ozone (2 ppm, *n* = 14) or filtered air (FA, control, *n* = 14) for 3 h, as described in [18]. Each group of exposed mice was either sacrificed at 4 h post-exposure (6 p.m., *n* = 12) or the following day at 11 a.m. (24 h, *n* = 16) using a ketamine/xylazine cocktail. Upon opening the peritoneal cavity, blood was obtained from the cava vein, and ovaries were collected and stored at −80 °C for processing as indicated below. All the procedures were approved by the Institutional Animal Care and Use Committee (IACUC) under protocol #46751.

### 2.2. Tissue Preparation and Gene Expression Measurements

At 4 h after exposure, RNA was extracted from one ovary (the other was used for histology), the pituitary gland, and the hypothalamus using Direct-zol RNA Miniprep Plus Kits (R2073, Zymo Research, Irvine, CA, USA) and then retro-transcribed to cDNA using the High-Capacity cDNA Reverse Transcription Kit (4368814, Applied Biosystems, Foster City, CA, USA). Taqman assays were used to determine the gene expression of gonadotropin-releasing hormone (GnRH1, Mm01315605_m1), luteinizing hormone (Lhb, Mm00656868_g1), follicle-stimulating hormone (Fshb, Mm00433361_m1), and their respective receptors (GnRHR, Mm00439143_m1; LHcgR, Mm00442931_m1; Fshr, Mm00442819_m1), aromatase (CYP19A1, Mm00484049_m1), and 17β-Hydroxysteroid dehydrogenase (HSD17B, Mm00501692_g1) using 18S (Mm03928990_g1) as an endogenous control through Real-Time PCR on a QuantStudio 3 (A28137, Applied Biosystems, Foster City, CA, USA). GnRH1 was measured in the hypothalamus; GnRHR, Lhb, and Fshb were measured in the pituitary; and Lhcgr, Fshr, CYP19A1, and HSD17B were measured in the ovaries. Relative gene expression was calculated using the 2^−ΔΔCT^ method. 

### 2.3. Ovarian Tissue Histology and Follicle Counts

Ovaries harvested at 4 h or 24 h after exposure were fixed and stained with H&E for histological analysis to determine the number of preovulatory follicles and corpora lutea. These slides were scanned using a Motic EasyScan Pro 6 and the Easy Scan software (Motic, Hong Kong, China). Ovary follicle counts with less than 180 or more than 400 total follicles were excluded from follicle analysis.

### 2.4. Serum Hormone Measurements

Serum from mice was separated from whole blood by centrifuging at 1300 rpm for 15 minutes. Serum from the 4 h timepoint was used to measure circulating estradiol, progesterone, and LH concentrations. All hormones were measured by ELISA (estradiol—MBS261250; progesterone—MBS266675, luteinizing hormone—MSB2018848).

### 2.5. Statistical Analysis

Dot plots were used to display follicle counts for stages of development as proportions of total follicles counted for each mouse. Differences between each follicle stage count and total follicle counts were determined using Student’s *t*-test. To determine differences in sex hormone concentration and relative gene expression between exposure groups, Student’s *t*-tests were also used. For gene expression statistical analysis, data were first log_2_-transformed. For all tests, a *p*-value of less than 0.05 was considered significant.

## 3. Results

### 3.1. Ozone Exposure Increases Serum Estradiol Levels

At 4 h after exposure, circulating progesterone levels were slightly increased in mice exposed to ozone vs. those exposed to filtered air, but this increase was not statistically significant (Figure 1A, *p* > 0.05). In contrast, the serum estradiol concentration was significantly increased in mice exposed to ozone compared to those exposed to filtered air (Figure 1B, *p* < 0.05). On average, the acute exposure to ozone resulted in a 30.1% increase in estradiol levels, when compared to female mice exposed to filtered air. On the other hand, LH serum levels were no different between ozone-exposed mice and filtered-air-exposed mice (Figure 1C, *p* > 0.05).

### 3.2. Ozone Exposure Does Not Alter GnRH Expression in the Hypothalamus but Affects Gonadotropin Receptor Expression in the Pituitary Gland

At 4 h after exposure, the expression of GnRH in the hypothalamic tissue of mice exposed to ozone (vs. filtered air) was not significantly different (not pictured). However, significant differences in GnRH receptor and FSH mRNA expression (*p* < 0.05), and near-significant differences in LH expression (*p* = 0.0515), were observed in mice exposed to ozone compared to those exposed to filtered air after 4 h (Figure 2).

### 3.3. Ozone Exposure Does Not Affect Gonadotropin Receptor and Steroidogenesis-Related Gene Expression in the Ovary

In ovarian tissue, exposure to ozone did not affect the expression of gonadotropin receptors (LH and FSH receptor) nor the steroidogenesis-related genes aromatase or 17β-hydroxysteroid dehydrogenase (17HSD) at 4 h after exposure (Figure 3).

### 3.4. Ozone Exposure Does Not Affect Ovarian Follicle Counts

When comparing primordial, secondary, preovulatory, and atretic follicles in fixed ovarian tissue from mice exposed to ozone vs. filtered air, we found no statistical differences in individual follicle proportions nor total follicles per mouse at either 4 h (Table 1) or 24 h (Table 2). While we did not find significant differences in follicle counts, a slight but notable difference in primary and atretic follicles was evident at the 24 h timepoint. When comparing corpora lutea in ovaries from mice exposed to ozone vs. filtered air, no significant differences were found at either timepoint.

## 4. Discussion

In this study, we investigated whether acute exposure to an inhaled pollutant affected ovarian hormone secretion and follicle development and the expression of genes involved in the HPG axis, folliculogenesis, and steroidogenesis. We found that circulating estradiol levels were significantly higher and that gonadotropin-releasing hormone receptor, luteinizing hormone, and follicle-stimulating hormone mRNA expression were reduced in the pituitary gland in mice exposed to ozone compared to those exposed to filtered air during the evening of proestrus. However, no changes in hypothalamic GnRH mRNA expression or ovarian aromatase, 17βHSD, and follicle counts were found. Our results indicate that acute ozone exposure can affect ovarian function and the subsequent secretion of steroid hormones, as well as the expression of pituitary gonadotropin genes.

The predominant female hormone, estradiol, is typically synthesized in the ovarian preovulatory follicles by enzymes that convert cholesterol to androgens and then to estrogens through aromatase and 17βHSD. While the expression of these two enzymes in the ovarian tissue was not significantly affected in our experiments, it is possible that changes in their expression occurred at an earlier time. On the other hand, adipose tissue and immune cells also express these enzymes [19,20], and could represent additional sites of estradiol synthesis in ozone-exposed mice, which could explain the higher serum estradiol levels. We also did not observe changes in preovulatory follicles (the site of estradiol synthesis) or other follicles between groups. However, the maturation of a follicle takes 17–19 days [21], which may explain why no changes in follicle proportions were observed. 

In the anterior pituitary gland, estradiol suppresses gonadotropin gene expression by binding to the nuclear estrogen receptor and repressing the expression of LHb and FSHb via estrogen response elements in their promoter regions [22]. At 4 h after ozone exposure, we observed significant (or near significant) reductions in both genes, which could be a direct consequence of the increase in serum estradiol. Similarly, we observed a significant decrease in the expression of the GnRH receptor. While typically estradiol upregulates this receptor’s expression via positive feedback during the preovulatory surge, sustained high estradiol levels are known to cause GnRH receptor downregulation [23,24]. On the other hand, the hypothalamic expression of the GnRH gene was not affected, as expected by changes in estradiol levels [25], which could also be due to timing and concentration [26] or estrogen receptor expression levels [27]. 

Prior epidemiological studies have suggested that exposure to air pollution affects fertility in both men and women [28,29]. One study by Merklinger-Gruchala et al. reported a 33% increase in circulating estradiol in women exposed to PM_10_ and cigarette smoke [30]. In a controlled human exposure study, Du et al. identified changes in estrogen receptor signaling genes after exposure to ozone in blood samples taken 2 h after a 2 h ozone exposure [31]. Both particle and gaseous components of smog (e.g., ozone) have also been linked to multiple endocrine-disrupting effects [32], including the dysregulation of thyroid, pituitary, gonadal, and adrenal hormones. The present study demonstrated that a single short exposure to ozone alters serum estradiol levels and pituitary gonadotropin gene expression. While our data using a single exposure do not indicate evident changes in follicle counts and ovarian gene expression at 4 h or 24 h post exposure, it is possible that changes in these parameters could occur after several weeks. Moreover, while we could not identify the direct cause of the significant increase in estradiol, it is possible that this hormone surge could affect other physiological functions in the short and long term. 

While our findings did not provide sufficient evidence to support any singular mechanism for the increase in estradiol circulating levels upon exposure to ozone, we believe that systemic inflammation caused by ozone exposure could potentially reduce the capacity for clearance of estradiol from the body. This mechanism is typically mediated through several cytochrome p450 enzymes such as CYP1A1, CYP1B1, and CYP1A2 [33,34]. Interestingly, CYP1A1 has been found in the lungs of mice and humans and is induced in the presence of aryl hydrocarbons, through the aryl hydrocarbon receptor (AHR) and some other common air pollutants [35]. However, some studies have indicated that its gene expression is reduced after exposure to ozone [36]. The other cytochrome p450 enzymes have not been reported to change expression in the presence of ozone, but they are induced during inflammation [37,38]. Thus, we hypothesize that the ligands inducing these enzymes during inflammation could be competing with estradiol, resulting in higher estradiol concentrations due to the lack of metabolism.

Our study has several limitations. Conducting a single acute ozone exposure experiment and measuring relatively short outcomes may have meant that our experimental timeline may not have identified long-term physiological effects. Measuring only selected genes and not protein expression also limits our ability to determine the mechanisms associated with the observed changes. Finally, we only tested one concentration of ozone for a short exposure and analyzed results at two selected timepoints. Future studies expanding these studies should consider sub-chronic and chronic ozone exposures, as well as short- and long-term effects. Despite these limitations, we have shown that ozone toxicity expands beyond respiratory and cardiovascular effects and could affect reproductive function.

## 5. Conclusions

In conclusion, we found that acute exposure to ozone in female mice during the morning of proestrus significantly affects estradiol levels and pituitary gonadotropin gene expression after 4 h. Acute ozone exposure did not alter the total number or relative proportions of ovarian follicles, steroidogenesis gene expression, or the circulating levels of gonadotropins. This is one of a few studies that use an experimental approach to study the fertility effects seen in epidemiological studies. Future studies examining the effect of chronic ozone exposures are warranted to validate epidemiological findings and identify relevant mechanisms that could link air pollution exposures with health effects, including fertility. 

## Figures and Tables

**Figure 1 ijerph-22-00222-f001:**
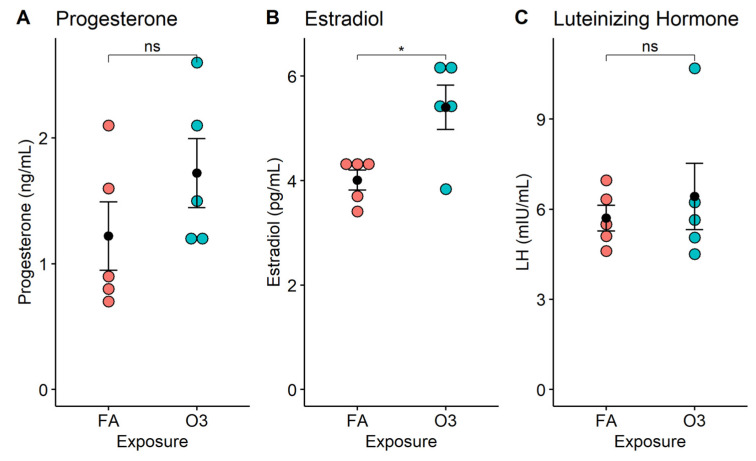
Serum hormone levels in female mice following acute exposure to ozone. Dot plots depict individual hormone concentrations (progesterone, estradiol, luteinizing hormone) in mouse serum at 4 h after exposure to 2 ppm ozone (O3) or filtered air (FA). *: *p* < 0.01; *n* = 5 mice per group; ns: not significant. (**A**) Progesterone; (**B**) Estradiol; (**C**) Luteinizing Hormone.

**Figure 2 ijerph-22-00222-f002:**
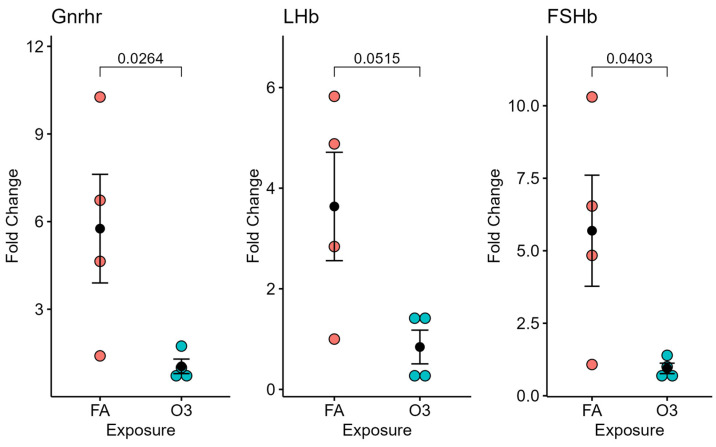
Pituitary-gland-selected gene expression at 4 h following ozone exposure. Relative gene expression of gonadotropin-releasing hormone receptor (Gnrhr), luteinizing hormone beta subunit (LHb), and follicle-stimulating hormone beta subunit (FSHb) in the pituitary gland (normalized to 18S) of female mice exposed to ozone (O3) or filtered air (FA); *n* = 4 mice per group.

**Figure 3 ijerph-22-00222-f003:**
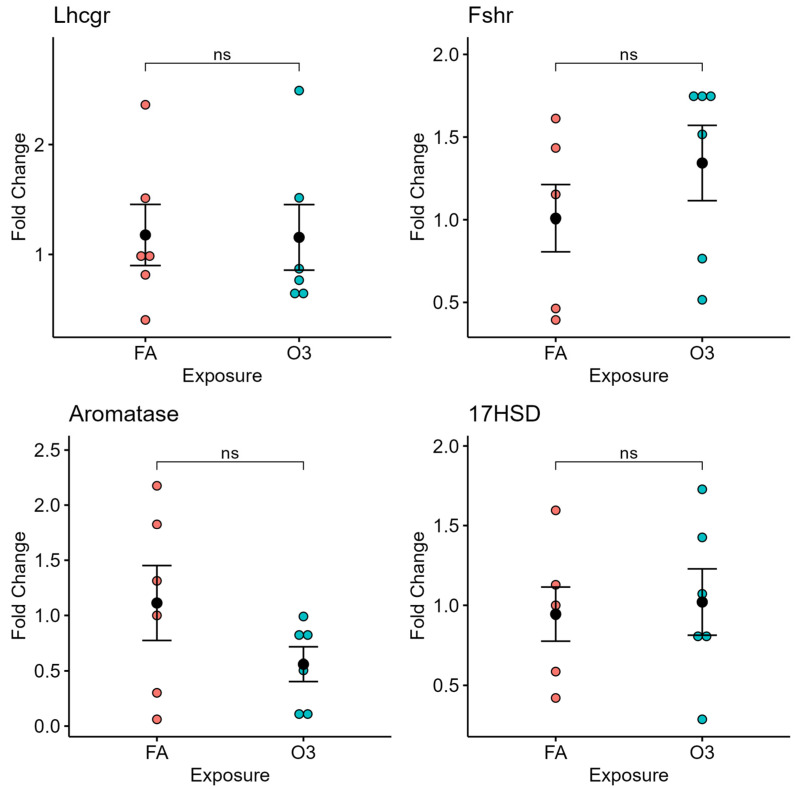
Ovarian gene expression at 4 h following ozone exposure. Relative gene expression of selected genes of luteinizing hormone receptor (Lhcgr), follicle-stimulating hormone receptor (Fshr), aromatase, and 17β-Hydroxysteroid dehydrogenase (17HSD) in ovarian tissue of mice exposed to ozone (O3) or filtered air (FA) at 4 h after exposure; *n* = 6 mice per group; ns: not significant.

**Table 1 ijerph-22-00222-t001:** Follicle counts at 4 h after exposure.

	Filtered Air (*n* = 5)	Ozone (*n* = 6)	
Follicle Stage	Average Count per Ovary [95% Confidence Interval]		*p*-Value
Primordial/Primary	66.4 [43.9, 88.9]	68.3 [50.3, 86.3)	0.933
Secondary	6.0 [2.4, 9.6]	7.7 [3.0, 12.3]	0.631
Preovulatory	9.6 [4.4, 14.8]	8.7 [5.5, 11.8]	0.745
Corpora Lutea	5.4 [3.0, 7.8]	4.8 [2.6, 7.1]	0.943
Atretic	154.8 [109.3, 200.3]	135.5 [103.8, 167.2]	0.632
Total	242.2 [190.5, 293.9]	225 [187.4, 262.6]	0.713

**Table 2 ijerph-22-00222-t002:** Follicle counts at 24 h after exposure.

	Filtered Air (*n* = 8)	Ozone (*n* = 7)	
Follicle Stage	Average Count per Ovary [95% Confidence Intervals]		*p*-Value
Primordial/Primary	49.8 [31.1, 68.4]	44.6 [37.1, 52.1]	0.849
Secondary	14.0 [8.8, 19.2]	17.6 [13.5, 21.6]	0.383
Preovulatory	33.3 [21.1, 45.4]	37.7 [34.1, 41.3]	0.924
Corpora Lutea	16.5 [10.8, 22.2]	20 [17.2, 22.8]	0.821
Atretic	144.6 [104.6, 184.6]	138.3 [118.2, 158.4]	0.426
Total	258.1 [215.4, 300.8]	258.1 [227.4, 288.9]	0.530

## Data Availability

All raw data are available on the Silveyra laboratory repository: https://psilveyra.github.io/silveyralab/ (accessed on 6 January 2025).

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
