# Peer review of "Acute Exposure to Ozone Affects Circulating Estradiol Levels and Gonadotropin Gene Expression in Female Mice"

_ijerph, 2025, doi:10.3390/ijerph22020222_

Round 1
Reviewer 1 Report
Comments and Suggestions for Authors
I read with care the article entitled “Acute exposure to ozone affects circulating estradiol levels and gonadotropin gene expression in female mice” from Rousselle and Silveyra. The cascade mechanism to explain ozone toxicity at respiratory level have been known for at least thirty years. However, the potential harmful impact of ozone at the level of fertility is more recent.
The experimental design adopted for this study traces earlier paper by one of the authors (ref. #13). However, the statistical analysis adopted and the representation of the results need more careful study. In detail: i) a summary table of the data obtained should be set up; ii) the figures should be reduced, eliminating those for which no statistically significant differences are highlighted.
Finally, it would be desirable to formulate a hypothesis on the cellular/nuclear/genetic mechanism that can explain how a contact between respiratory mucosa/blood circulation and ozone can lead to alterations in the gene expression of molecules that affect reproduction and the production of gonadotropins.
Author Response
I read with care the article entitled “Acute exposure to ozone affects circulating estradiol levels and gonadotropin gene expression in female mice” from Rousselle and Silveyra. The cascade mechanism to explain ozone toxicity at respiratory level have been known for at least thirty years. However, the potential harmful impact of ozone at the level of fertility is more recent.
We thank the reviewer for the comments and recommendations to improve our work.
The experimental design adopted for this study traces earlier paper by one of the authors (ref. #13). However, the statistical analysis adopted and the representation of the results need more careful study. In detail: i) a summary table of the data obtained should be set up; ii) the figures should be reduced, eliminating those for which no statistically significant differences are highlighted.
We have revised the manuscript accordingly and removed the figure depicting the gene expression of gonadotropin-releasing hormone in the hypothalamus. However, we believe that visual representations through figures are more straightforward to parse for the included data than a summary table. Additionally, we think it is essential that all data produced is shown, even when not statistically significant, to present the whole picture of the data obtained. Furthermore, since all gene expression data are part of the pathway of steroidogenesis of estrogens, having them all displayed provides a whole picture and potential further avenues of research. Moreover, reviewer 3 stated, “Generally, the figures and tables are important for adequately understanding the content presented,” therefore, we have kept most of the figures and tables that we considered relevant.
Finally, it would be desirable to formulate a hypothesis on the cellular/nuclear/genetic mechanism that can explain how a contact between respiratory mucosa/blood circulation and ozone can lead to alterations in the gene expression of molecules that affect reproduction and the production of gonadotropins.
We agree with the reviewer and have added more information to the discussion section, including potential mechanisms involved in our findings. We hope that these additions have improved the quality of our study.
Reviewer 2 Report
Comments and Suggestions for Authors
comments are added in the attached word file

the paper is well written
Author Response
Dear Author, the paper is a good quality paper it just needs some English improvements and the comments about the work are listed below
We thank the reviewer for the constructive feedback. We have modified the manuscript accordingly.
Abstract: page 1, line 11, can you add why you use 8-week-old female C57BL/6J mice as the animal model for this study, is it according to literature or according to what strategy? In addition what is the methods that the authors used to quantify changes in follicle development after ozone exposure.
We thank the reviewer for the recommendation; we added a few words to the abstract but kept most of the details in the methods section due to character limit constraints.
Introduction, you need to add more details about the same work form literature as it is very shallow showing only general theoretical background.
Unfortunately, the only work conducted on this topic is through epidemiological studies; no experimental/animal studies address the relationship between air pollution exposures and reproductive function. We believe our work is important to raise awareness and as a starting point for future studies examining mechanisms, and we have emphasized this in the revised version.
Line 57, add more about the aim of your study.
Line 64, from where your purchased the mice and add more details please.
Thank you for the suggestions; we have added additional information in both sections.
Line 69 what are the procedure you did (Blood and ovaries were collected at the time of sacrifice. All the procedures were approved by the Institutional Animal Care and Use Committee (IACUC) under protocol #46751) add details.
Additional details have been provided.
Line 111, why you wrote that (levels were not significantly different between groups (Figure 1A and C).) please discuss it properly.
The statistical analysis of the results in Figures 1A and 1C revealed no significant differences across groups. We have edited the language of this statement for clarity.
Line 151 (While differences in follicle counts were not significant, a slight but no table difference in primary and atretic follicles was evident at the 24 h timepoint. Similarly, no significant differences in corpora lutea were found at either timepoint.) please link it with literature results to show the difference.
This statement refers to our results, which are presented in the tables from our work and not the literature. We have modified this sentence for clarity.
Make result and discussion together please so it can be more obvious.
Unfortunately, combining results and discussion will violate the journal style guidelines. To help the reader, we have repeated relevant results in the discussion section to improve clarity.
Add more recent references from the same journal you are going to publish that is very relevant to your work.
Thank you. While we have considered this request, the references available (including the ones listed below) are not relevant to the work presented.
Murtadah, I., AI-Sharify, Z.T., Hasan, M.B. Atmospheric concentration saturated and aromatic hydrocarbons around dura refinery (2020) IOP Conference Series: Materials Science and Engineering, 870 (1), art. no. 012033. https://doi.10.1088/1757-899X/870/1/012033
This paper is irrelevant to the stated work. The term ozone appears once within the paper as a reference to the fact that hydrocarbons are part of the atmospheric chemistry that can produce ozone.
Please link between ozone exposure and the observed reduction in gene expression? And please prove what is What mechanisms are proposed to explain the significant decrease in the expression of the GnRH receptor following ozone exposure?
Line 210, add more to the conclusion it is very shallow please add more details.
Thank you for the suggestions to discuss mechanisms relating exposures and gene expression changes and expanding the conclusion. We have modified this section to expand the discussion and speculate potential mechanisms based on our data and the literature.
Reviewer 3 Report
Comments and Suggestions for Authors
The article starts from the assertion that ‘ozone, a critical air pollutant, has been shown to lead to systemic inflammation that can alter bodily functions, including hormone secretion and fertility, and the hypothalamic-pituitary-gonadal (HPG) axis.’ Thus, the aim of the study was ‘to quantify changes in hormone production and follicle development after acute exposure to ozone using an animal model.’ In alignment with the journal’s scope and relevant to the field of knowledge, the topic reveals potential interest for its readers and the manuscript has traits of originality, despite the non-definitive stage of the results achieved, prompting the need for complementary studies to effectively fill gaps in the area and provide substantial additions compared to other published materials.
Both in the abstract and the introductory part of the text, there could be greater clarity about the main question addressed by the research, although this is explicitly stated in the latter section with the hypothesis that ‘acute ozone exposure alters the development of ovarian follicles and affects ovarian hormone secretion.’ As for Section 1 (Introduction), it is noticeable that the contexts, concepts, and justifications are based on more than 58% of references that are over five years old, suggesting the possibility of updating the state-of-the-art of the topic.
In Section 2 (Materials and Methods), an important clarification is provided that ‘all the procedures were approved by the Institutional Animal Care and Use Committee (IACUC) under protocol #46751’, but the methodological approaches could be more detailed, for example, with specific quantities of mice analyzed in each group and the experimentation date.
Section 3 (Results) succinctly and directly demonstrates that ozone exposure increases serum estradiol levels (Subsection 3.1), does not alter GnRH expression in the hypothalamus but affects gonadotropin receptor expression in the pituitary gland (Subsection 3.2). On the Other hand, it does not affect gonadotropin receptor and steroidogenesis-related gene expression in the ovary (Subsection 3.3) and ovarian follicle counts (Subsection 3.4).
The discussion, presented in Section 4, engages in an interesting debate with the literature. However, there is not a strong alignment with the introductory references, and the cited sources account for just over 28% from the last five years and nearly 57% from the last decade (including the previous ones).
Although Section 5 (Conclusions) is consistent with the evidence presented and addresses the hypothesis posed, it is very succinct. Thus, it is recommended that the study's limitations and its effective contributions to filling gaps in the area and adding compared to other published materials be clearly outlined. It is also suggested to include details regarding the conduct of future studies.
Overall, the references are appropriate, but they should be more up-to-date to ensure the state-of-the-art relevance of the theme. Only about 33% are from the last five years, and just under 60% are from the last decade. Generally, the figures and tables are important for adequately understanding the content presented.
Author Response
The article starts from the assertion that ‘ozone, a critical air pollutant, has been shown to lead to systemic inflammation that can alter bodily functions, including hormone secretion and fertility, and the hypothalamic-pituitary-gonadal (HPG) axis.’ Thus, the aim of the study was ‘to quantify changes in hormone production and follicle development after acute exposure to ozone using an animal model.’ In alignment with the journal’s scope and relevant to the field of knowledge, the topic reveals potential interest for its readers and the manuscript has traits of originality, despite the non-definitive stage of the results achieved, prompting the need for complementary studies to effectively fill gaps in the area and provide substantial additions compared to other published materials.
We thank the reviewer for the comments and recommendations to improve our work.
Both in the abstract and the introductory part of the text, there could be greater clarity about the main question addressed by the research, although this is explicitly stated in the latter section with the hypothesis that ‘acute ozone exposure alters the development of ovarian follicles and affects ovarian hormone secretion.’ As for Section 1 (Introduction), it is noticeable that the contexts, concepts, and justifications are based on more than 58% of references that are over five years old, suggesting the possibility of updating the state-of-the-art of the topic.
We thank the reviewer for these suggestions. We have modified the abstract to make the aim of the study clearer to the reader. Regarding references, many of the available sources we used highlight core concepts of biochemistry and known pathways in human physiology, thus appearing “out-of-date.” A PubMed search of "ozone" and "ovaries" for the past 5 years only yielded nine results. While we have tried to add newer references, our work represents an understudied and primarily unexplored area.
In Section 2 (Materials and Methods), an important clarification is provided that ‘all the procedures were approved by the Institutional Animal Care and Use Committee (IACUC) under protocol #46751’, but the methodological approaches could be more detailed, for example, with specific quantities of mice analyzed in each group and the experimentation date.
We have updated this section to add the numbers of the experimental mice.
Section 3 (Results) succinctly and directly demonstrates that ozone exposure increases serum estradiol levels (Subsection 3.1), does not alter GnRH expression in the hypothalamus but affects gonadotropin receptor expression in the pituitary gland (Subsection 3.2). On the Other hand, it does not affect gonadotropin receptor and steroidogenesis-related gene expression in the ovary (Subsection 3.3) and ovarian follicle counts (Subsection 3.4).
Thank you for highlighting our main results. We have modified the text in this section to improve clarity.
The discussion, presented in Section 4, engages in an interesting debate with the literature. However, there is not a strong alignment with the introductory references, and the cited sources account for just over 28% from the last five years and nearly 57% from the last decade (including the previous ones).
We have added references to this section, although these are limited, as stated above.
Although Section 5 (Conclusions) is consistent with the evidence presented and addresses the hypothesis posed, it is very succinct. Thus, it is recommended that the study's limitations and its effective contributions to filling gaps in the area and adding compared to other published materials be clearly outlined. It is also suggested to include details regarding the conduct of future studies.
We appreciate the suggestions. We have modified the manuscript to clarify how our study addresses gaps in the field, and discussed future directions.
Overall, the references are appropriate, but they should be more up-to-date to ensure the state-of-the-art relevance of the theme. Only about 33% are from the last five years, and just under 60% are from the last decade. Generally, the figures and tables are important for adequately understanding the content presented.
Thank you for this comment. As mentioned earlier, work in this field is limited, and thus recent references are scarce. We hope our work helps bring awareness to this topic and encourages more studies that could help understand the effects of environmental exposures on endocrine function.